# Criterion and Construct Validity of the Pocket-Worn RISE Device to Assess Movement Behaviour in Community-Dwelling People with Stroke

**DOI:** 10.3390/s25113308

**Published:** 2025-05-24

**Authors:** Camille F. M. Biemans, Laura van der Heiden, Cindy Veenhof, Olaf W. Verschuren, Johanna M. A. Visser-Meily, Martijn F. Pisters, Yvonne A. W. Hartman

**Affiliations:** 1Department of Rehabilitation, Physiotherapy Science & Sport, UMC Utrecht Brain Centre, University Medical Centre Utrecht, Utrecht University, 3584 CXUtrecht, The Netherlands; 2Research Group Empowering Healthy Behaviour, Department of Health Innovations and Technology, School of Health Sciences, Fontys University of Applied Sciences Physiotherapy, 5612 MA Eindhoven, The Netherlands; 3Centre for Physiotherapy Research and Innovation in Primary Care, Julius Health Care Centres, 3541 CD Utrecht, The Netherlands; 4Research Group Innovation of Human Movement Care, HU University of Applied Sciences, 3584 CS Utrecht, The Netherlands; 5Center of Excellence for Rehabilitation Medicine, UMC Utrecht Brain Center and De Hoogstraat Rehabilitation, 3583 TM Utrecht, The Netherlands

**Keywords:** wearable electronic devices, validity, sedentary behaviour, physical activity, accelerometry

## Abstract

Accurate monitoring of physical activity (PA) and sedentary behaviour (SB) is crucial for tailoring interventions aimed at improving movement behaviour. This study evaluated the validity of the pocket-worn RISE device for measuring movement behaviour in community-dwelling patients with stroke. Criterion validity was assessed in a laboratory setting using video recordings. Construct validity was assessed in a free-living setting using the thigh-affixed ActivPAL. In the laboratory setting (25 participants, 66 ± 11.8 years), the RISE device showed good criterion validity for SB and PA. In the free-living setting (19 participants, 73 ± 10.2 years), the RISE device showed good construct validity for SB and PA, though further improvements could enhance the accuracy of individual-level measurements of moderate to vigorous physical activity (MVPA) and prolonged sedentary bouts. The mean absolute percentage error and mean percentage error were below the predefined 20% threshold for SB and PA. Intraclass correlation coefficients (ICCs) for SB and PA showed good reliability, but ICC ranges for prolonged sedentary bouts and MVPA were too broad to draw firm conclusions. These findings indicate the RISE device is well-suited for measuring SB and PA in free-living conditions. With real-time feedback, app compatibility, and pocket-wear convenience, the RISE device shows potential for behavioural interventions targeting movement behaviour in stroke and other chronic conditions.

## 1. Introduction

The growing stroke incidence, combined with improvements in acute care, has led to a higher prevalence of people living with limited residual symptoms after stroke [1]. Around 90% of the stroke population is living at home, known as community-dwelling people. These people are at higher risk of a new major cardiovascular event (MACE) or death. One of the known risk factors for MACE is excessive sedentary behaviour (SB) and low physical activity (PA). SB is defined as “any waking activity characterised by an energy expenditure of ≤1.5 metabolic equivalents (METs) while in a sitting or reclining posture” [2]. Studies have shown that the stroke population spends about 10–14 h per day in SB with minimal engagement in moderate to vigorous PA (MVPA) [3,4]. This highlights the need for behaviour-oriented interventions focusing on reducing SB by supporting PA.

An essential part of such an intervention is objectively and accurately monitoring SB and PA. Accelerometers are preferred over subjective measurements for their precision and accuracy. They help mitigate challenges associated with recall bias and social desirability [5]. Some accelerometers can also serve as intervention tools by providing insights and real-time feedback about someone’s SB and PA [6]. The ActivPAL accelerometer is widely considered the gold standard for monitoring movement behaviour in free-living settings among individuals with stroke, as it accurately differentiates between various postures and activities [7]. However, it primarily serves as a research device. Consumer-based wearables are preferred over research devices for coaching due to their practicality, app compatibility, and ability to provide real-time feedback [8].

Recognising the need for a device that offers both monitoring and real-time feedback, the RISE (Reduce and Interrupt sedentary behaviour using a blended behavioural intervention to Empower people at risk towards sustainable 24-h movement behaviour change) study utilises the pocket-worn RISE device. This is an Activ8 (Wearable Activity Monitor Generation 2) customised to align with the RISE intervention, featuring a vibration function and compatibility with the RISE eCoaching system [9,10]. Previous studies have validated the Activ8 for measuring activities in hospitalised patients and patients with stroke-related mobility impairments [5,11]. In this study, we are interested in valid activity monitoring for community-dwelling people with stroke. The Activ8 is preferred over other consumer-based and research monitors since it is cost-friendly, pocket-worn, waterproof, compatible with apps, and can continuously monitor for up to 30 days without charging.

Compared to research devices, coaching devices exhibit a lower accuracy for monitoring movement behaviour due to individual, environmental and contextual factors that make it challenging to measure [8]. Although the validity of coaching devices may be less rigorous than research-grade tools, an acceptable accuracy of coaching devices is still crucial for providing effective real-time feedback in free-living settings. Accurate devices can play a valuable role in monitoring movement behaviour in interventions focusing on sustainable movement behaviour change. Therefore, this study aims to assess the validity of the pocket-worn RISE device in laboratory and free-living settings among community-dwelling people with stroke.

## 2. Materials and Methods

### 2.1. Study Design

This cross-sectional study combined both laboratory and free-living settings to evaluate the RISE device for construct and criterion validity separately (Figure 1) [12]. The validity of the pocket-worn RISE device was measured with camera recordings and the thigh-affixed ActivPAL as reference standards.

### 2.2. Participants

Participants were recruited from the Department of Neurology of University Medical Centre Utrecht, Centre for Geriatric Rehabilitation De Parkgraaf Utrecht, the primary care physiotherapy practice Vita Forum Bakel, and the primary care physiotherapy practice Fysiocompany, all based in the Netherlands. Inclusion criteria were clinically confirmed stroke, aged 18 and older, capable of independently performing daily activities, and able to speak Dutch. The exclusion criterion was severely affected cognitive ability based on the opinion of the participant’s physiotherapist. Participants were asked to wear lower-body clothing with front pockets during measurements. A target sample size of 20–30 participants for each validity assessment was determined based on a review of similar studies on accelerometer validity [13]. All participants provided written informed consent prior to participation, in accordance with ethical guidelines. This study was conducted in accordance with the Declaration of Helsinki and was approved by the Medical Ethics Committee of University Medical Centre Utrecht, The Netherlands (15-768/C).

### 2.3. Measurement Instruments

The RISE device is an Activ8 accelerometer (Wearable Activity Monitor Generation 2) with a vibration function and is compatible with the RISE eCoaching system [9,10]. Although the RISE device includes a vibration function, this feature was not used in the current validation study and did not influence data analysis. The Activ8 (trademark of Remedy Distributions Ltd., 2M Engineering, Valkenswaard, The Netherlands) is a small (30 × 32 × 10 mm), lightweight (20 g) triaxial accelerometer. The RISE device contains a battery, a real-time clock, and a medium for data storage. The RISE device records data in 5 s bouts. Per bout, it registers one of the following categories: lying, sitting, standing, walking, cycling, and running. It is worn in the front pocket of lower-body clothing on the non-affected side.

#### 2.3.1. Laboratory Setting

Video footage is considered the gold standard for assessing the criterion validity of accelerometers in laboratory settings [12]. A video camera (JVC Camcorder GZ-R495BE) was mounted on a tripod to record activities during the protocol.

#### 2.3.2. Free-Living Setting

ActivPAL (PAL Technologies Ltd., Glasgow, UK) is considered the gold standard for assessing the construct validity of accelerometers in free-living settings [7]. ActivPAL incorporates a triaxial accelerometer, and movement data were recorded in bouts of one second. Per bout, ActivPAL detects one of the following categories based on PALanalysis software (Version 8.11.8.75, PAL Technologies Ltd., Glasgow, UK) with the CREA algorithm: primary lying, secondary lying, non-wear, sedentary time, seated transport time, standing, walking, and cycling. ActivPAL is affixed to the thigh in a waterproof sleeve with hypoallergenic tape.

### 2.4. Measurement Procedure

Participant characteristics were obtained from medical records: type of stroke, stroke location, time since stroke, age, and sex. Further participant characteristic measurements were height, weight, functional independence (Barthel Index [14]), walking ability (Functional Ambulation Category [15]), and comfortable walking speed (10-metre walking test [16]).

#### 2.4.1. Laboratory Setting

To assess criterion validity, participants completed a 19 min testing protocol (Table A1) while being filmed as a reference standard. The lying task was extended to seven minutes to meet the RISE device’s threshold for detecting non-wear, defined as a period of signal absence exceeding five minutes. Because the device does not explicitly classify ‘lying’, any detected non-wear during this controlled task was interpreted as ‘lying’, as continuous device wear was ensured throughout the protocol. All other activities were timed at 90 s each, following protocols used in other accelerometer validation studies [5,17]. If participants were physically unable to perform an activity, the activity was skipped. The camera and RISE device were time-matched by starting the camera simultaneously with the RISE device. The camera clock was synchronised with the laptop used to configure the RISE device.

#### 2.4.2. Free-Living Setting

To assess construct validity, participants wore the pocket-worn RISE device and thigh-affixed ActivPAL during waking hours for two consecutive days. This approach is in line with current activity monitor validation guidelines, which recommend collecting at least 24 h of free-living data to ensure the ecological validity of the measurements [12]. Both devices were initialised simultaneously using the same laptop to ensure synchronised start times. Participants were instructed to wear both devices during all waking hours and to log their wear times in a provided logbook. A valid measurement day was defined as a day with at least 10 h of continuous data output [18]. Days with less than 10 h of wear time were excluded from the analysis. Participants were included in the analysis if they had at least one valid measurement day (≥10 h of data). After the measurement period, participants returned the devices by mail.

### 2.5. Data Analysis

Statistical analyses were performed in SPSS (Version 30.0.0, IBM Corp., Armonk, NY, USA) and Microsoft Office Excel 2021 (Microsoft Corporation, Redmond, WA, USA). Descriptive statistics were used to describe participant characteristics.

#### 2.5.1. Laboratory Setting

To assess criterion validity, the data output from the RISE device was processed using a conversion tool (Activ8 GEN2 Pro PC Application, 2M Engineering, Valkenswaard, The Netherlands). No manual settings or additional criteria were applied during this process. Categories (lying, sitting, standing, walking, cycling) from video recordings were taken as criterion measures. Video recordings were synchronised with protocol completion using a time-stamped event at the start of the recording. These annotations served as the reference standard against which the RISE output was validated. Disagreements were resolved through slow-motion video review and consensus between two independent observers [12]. The middle 60 s portions of 90 s recordings were analysed. For ‘lying’, the middle 60 s portion of the 120 s recording was used.

Video footage and RISE device data were compared per 5 s bout. Validation categories were SB and PA. SB consisted of lying and sitting. PA consisted of standing, walking, and cycling. Sensitivity, specificity, and positive predictive value (PPV) were calculated for SB and PA. Scores below 0.60 indicate poor sensitivity or specificity; scores between 0.60 and 0.75 indicate moderate sensitivity or specificity; and scores between 0.75 and 1.00 indicate good sensitivity or specificity [19]. For group insights, 95% confidence intervals were calculated. Misclassification rates were analysed to identify categorisation errors. A secondary analysis evaluated walking consistency at different speeds (2, 3, 4, 5 km/h) on a treadmill, focusing on sensitivity scores for detecting walking activities at varying speeds.

#### 2.5.2. Free-Living Setting

To assess construct validity, the data output from ActivPAL was processed using PALanalysis software (Version 8.11.8.75, PAL Technologies Ltd., Glasgow, UK) with the CREA algorithm. No manual settings or additional criteria were applied during this process. ActivPAL categories were primary lying, secondary lying, non-wear, sedentary time, seated transport time, standing, walking, and cycling. The RISE device categories were lying, sitting, standing, walking and cycling. Segments of non-wear time as indicated in the logbook were manually excluded from the dataset for RISE device and ActivPAL. In all other cases, periods detected as non-wear by the RISE device were retained and labelled as ‘lying’, since continuous device wear was assumed during these periods.

RISE device and ActivPAL data were segmented into 5 s bouts for further analysis. Subsequently, categorised data from ActivPAL were compared with corresponding categories from the RISE device. The validation categories were in line with the RISE eCoaching system [9,10]: SB: lying and sitting (RISE device), primary lying, secondary lying, sedentary time, and seated transport time (ActivPAL); prolonged sedentary bouts: continuous SB periods lasting longer than 30 min; PA: standing, walking, and cycling (RISE device), upright time, stepping time, and cycling time (ActivPAL); MVPA: activities with an estimated METs value >3, based on standard conversion guidelines [2]. MVPA was categorised using the conversion tool (2M Engineering (RISE device) and the script of Winkler et al. (ActivPAL) [20]. No manual settings or additional criteria were applied during this process.

The normality of measurements was assessed using Shapiro–Wilk tests. Mean absolute percentage error (MAPE) was used to assess individual agreement, as recommended in an expert statement and review [12,13]. The recommended criterion for MAPE for clinical purposes in free-living settings is ≤20% for evaluating agreement, acknowledging the balance between stringency and practical application in free-living settings [8,21]. Values closer to zero indicate higher accuracy. For example, a MAPE of 20% implies that, on average, the RISE device values are 20% off from ActivPAL values. Mean percentage error (MPE) was used to assess group agreement, evaluating overall bias and directionality in errors between the RISE device and ActivPAL [12,13]. An MPE of ≤20% was selected as an acceptable criterion [8,21]. Values closer to zero indicate higher accuracy. For example: an MPE of 20% implies that, on average, RISE device values are 20% off from ActivPAL values while considering the direction of these deviations. MAPE and MPE cannot be used when actual values are close to zero, as this will cause division by zero or extremely high error values, leading to misleading accuracy results [22]. Therefore, MAPE and MPE were not computed when any value was <1.0 [22].

The Bland–Altman method and limits of agreement (LoA) were used to understand the nature and source of error [13]. Plots illustrate the mean difference (ActivPAL minus RISE device) measurements against the average of both measurements. LoA show 95% confidence intervals for individual observations. Mean differences close to zero signify a more accurate device. Intraclass correlation coefficients (ICCs) with 95% confidence intervals were used to assess the level of agreement between the RISE device and ActivPAL measurements using a two-way mixed effects model with absolute agreement [23]. ICC values of >0.9, ≤0.9 to >0.8, ≤0.8 to >0.5, and ≤0.5 were considered as excellent, good, moderate, and poor reliability [23].

## 3. Results

### 3.1. Participants

This study included 39 participants: 12 participated in the laboratory setting, 13 participated in the free-living setting, and 14 participated in both. One participant was excluded from the laboratory setting due to a malfunctioning device. Eight participants were excluded from the free-living setting after applying the wear time criterion of more than 10 h per day. Ultimately, 25 participants were included for analysis of the laboratory setting and 19 for the free-living setting. Participant characteristics are summarised in Table 1.

### 3.2. Laboratory Setting

Criterion validity was assessed using data from 25 participants. The RISE device showed sensitivity scores of 0.93 and 0.96 for SB and PA, respectively. Specificity scores for SB and PA were 0.96 and 0.93. The PPV for SB and PA was 0.95 (Table 2). Not all participants completed the entire protocol due to physical limitations. The number of observations for each protocol component can be found in Appendix A.

SB was classified as PA 6.6% of the time. If so, it was registered as standing (52.8%), walking (45.3%), or cycling (1.9%). PA was misinterpreted as SB 1.3% of the time. If so, it was interpreted as lying (5.0%) or sitting (95.00%) (Table 3). The RISE device showed a sensitivity [95% confidence interval] of 0.92 [0.89–0.94] for walking at 2 km/h, 1.00 [1.00–1.00] for 3 km/h, 0.94 [0.91–0.97] for 4 km/h, and 0.91 [0.86–0.96] for 5 km/h.

### 3.3. Free-Living Setting

Construct validity was assessed using data from 19 participants. Data from one participant were based on single-day recordings due to insufficient data (<10 h) on the second day. All movement categories demonstrated a normal distribution, except for MVPA data from the RISE device. The MAPE for SB and prolonged sedentary bouts was 9.7 and 19.8, respectively. The MPE for SB and prolonged sedentary bouts was 11.0 and 1.6, respectively (Table 4). MAPE and MPE could not be analysed for PA and MVPA since ≥1 value was close to zero [21]. For SB, prolonged sedentary bouts, PA, and MVPA, the ICCs [95% confidence intervals] were 0.8 [0.5–0.9], 0.7 [0.2–0.9], 0.8 [0.5–0.9], and 0.5 [−0.2–0.8], respectively (Table 4).

Bland–Altman plots revealed the following mean differences over two measurement days: 0 h and 57.6 min for sedentary behaviour, 2 h and 7.2 min for time spent in prolonged sedentary bouts, −0 h and 27 min for physical activity, and 0 h and 57.6 min for MVPA (Figure 2).

## 4. Discussion

This study evaluated the validity of the pocket-worn RISE device (Activ8) for monitoring sedentary behaviour (SB) and physical activity (PA) in laboratory and free-living settings in community-dwelling people with stroke. Gold standards were used as reference methods, using video recordings and ActivPAL thigh-affixed research devices. In the laboratory setting, the RISE device demonstrated good criterion validity for SB and PA. In the free-living setting, construct validity showed high agreement between the RISE device and ActivPAL for measuring SB and PA, though further improvements could enhance the accuracy of individual-level measurements of moderate to vigorous physical activity (MVPA) and prolonged sedentary bouts. The ICCs for SB and PA showed good reliability, while ICC ranges for prolonged sedentary bouts and MVPA were too broad to draw firm conclusions. The findings of this study indicate that the pocket-worn RISE device is well suited for measuring SB and PA in free-living settings, highlighting its potential for use in behavioural interventions targeting movement behaviour.

### 4.1. Laboratory Setting

Criterion validity assessment provided insight into the error margins of the RISE device, which appeared to be highly acceptable. The RISE device exhibited a strong ability to differentiate between SB and PA in the laboratory setting, with good sensitivity and specificity values. Misclassifications were limited, with SB incorrectly classified in only 6.6% of cases and PA in 1.3% of cases. Previous studies showed similar results in the Activ8 validity of movement behaviour in patients with stroke-related mobility impairments (with a similar laboratory setting) and hospitalised patients [5,11]. The established criterion validity in a controlled setting provides a necessary foundation for reliable use in free-living settings where movement patterns are more variable.

As the accuracy of accelerometers is prone to decrease when gait speed and step frequency decrease [24,25], we tested the validity of the RISE device at multiple walking speeds. The RISE device showed good validity across different walking speeds. There was no clear relationship observed between walking speed and device accuracy. This strengthens the idea of using the RISE device for measuring movement behaviour in individuals with stroke or other chronic conditions that result in slower walking speeds.

Given that not all participants completed the highest treadmill speed condition, and acknowledging that lower functional capacity may influence movement patterns and measurement outcomes, a sensitivity analysis was conducted. Analyses were repeated in the subset of participants who completed the 5 km/h condition (N = 11) to assess the potential impact of excluding participants with lower functional capacity. The results remained consistent, indicating that the findings were not biased by differences in functional capacity at higher walking speeds.

### 4.2. Free-Living Setting

Construct validity assessment demonstrated that the accuracy (MAPE) for SB and prolonged sedentary bouts was within an acceptable range for practical use [8]. Since the RISE device is used on an individual basis, MAPE analysis is particularly relevant, as it reflects accuracy at the individual level [9,10]. These results ensure the RISE device’s suitability for personalised coaching on sedentary behaviour in a free-living setting. On the group level, the Bland–Altman plots for SB and PA demonstrated a mean difference close to zero (57 and 27 min over two measurement days, respectively). This indicates that discrepancies between the devices are at most one hour on the group level. However, the plots reveal greater variation in SB at the individual level. This suggests that individual measurements may sometimes deviate more from actual values. Only one study has examined Activ8’s accuracy in a free-living setting among patients with stroke and found similar results. However, this study had people perform pre-set protocolised ‘free-living’ activities rather than activities of their choice, and patients were more physically impaired than our study population [5].

An increasing number of individuals track their movement behaviour in free-living settings using consumer-grade wearables. A systematic review of these devices found no equivalence between consumer-grade and research-grade PA monitors for assessing time spent in SB [26]. While Fitbit Flex seemed reasonably accurate for steps and intensity, it was less reliable for MET estimation and showed discrepancies between day-level and minute-level measurements [27]. Its lack of transparency and frequent software updates raise concerns about its data reliability. In contrast, the RISE device showed strong accuracy for SB and PA, making it a more reliable option for free-living interventions.

The RISE device agreement seemed good for measuring SB and PA but demonstrated moderate to poor agreement for prolonged sedentary bouts and MVPA. Wide ranges in the ICC and 95% CI of the Bland–Altman plots indicate substantial variability in its measurements. For example, the wide range in ICC for MVPA suggests that, with 95% confidence, the true value lies between −0.2 and 0.9, making it impossible to draw reliable conclusions about the device’s consistency for measuring MVPA. This suggests that additional data may be required to obtain a more precise estimate, although the recommended sample size for validation studies on accelerometers was met [13]. The RISE device generally underestimates prolonged sedentary bouts compared to ActivPAL, suggesting that prolonged sedentary bouts are more accurately captured by the thigh-affixed ActivPAL. A contributing factor might be the pocket placement of the RISE device, which makes it more susceptible to movement artefacts. As a triaxial accelerometer, a shift in orientation in the pocket can lead to variability in measured acceleration. Any slight movement within the 30 min sitting period may disrupt the classification of prolonged sedentary bouts. In contrast, the thigh-affixed ActivPAL is not prone to such disturbances. Nevertheless, the intended use of the RISE device is for coaching in SB and PA in a free-living setting. With strong agreement on SB and PA, the RISE device appears to be well-suited for this purpose.

### 4.3. Strengths and Limitations

A strength of this study is that both criterion and construct validity were investigated using robust reference methods to enhance reliability. All analyses are in line with a recommended expert statement for assessing the validity of consumer wearables [12]. A limitation is the low ecological validity of laboratory setting measurements. Activity periods were longer than those typically seen in free-living settings, and transition times were included in the analysis, unlike in free-living settings. Fanchamps et al. found similar agreement between laboratory and free-living settings, despite not accounting for transition time [5]. This suggests that transition time may not impact the results.

Importantly, Fanchamps et al. recommended that future research enhance ecological validity by allowing participants to self-select activities in a free-living environment, rather than adhering to a predefined activity protocol [5]. This approach was implemented in the present study. To the best of our knowledge, this is the first study to apply this recommendation specifically within a stroke population.

As aforementioned, wearing the RISE device in the pocket may have resulted in reduced accuracy due to variability in pocket sizes and types of pants and accompanying excessive movement. Prior research suggests that direct attachment to the thigh could lead to higher validity [28,29]. Attaching the monitor to the frontolateral thigh has been shown to result in better body detection compared to pocket placement [28,29]. However, despite not being firmly attached, the device demonstrated reliability within acceptable ranges in our study, supporting its suitability for free-living settings. Furthermore, comfort in wearing the device promotes adherence, which is essential for achieving lasting behaviour change. A limitation of this study is the absence of intensity level validation in the laboratory setting, for example, by using a metabolic cart to validate physical activity intensity classifications.

In free-living settings, the pocket-worn RISE device seemed well-suited for measuring SB and PA, though further improvements could enhance the accuracy of individual-level measurements of MVPA and prolonged sedentary bouts. Together with its ability to provide real-time feedback, app compatibility, and the convenience of pocket-wear, the RISE device shows potential for behavioural interventions targeting movement behaviour in stroke and other chronic conditions.

## Figures and Tables

**Figure 1 sensors-25-03308-f001:**
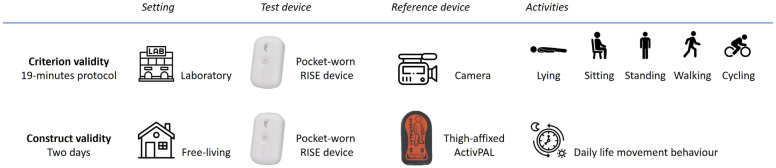
Assessment procedure of criterion and construct validity in laboratory and free-living settings, respectively.

**Figure 2 sensors-25-03308-f002:**
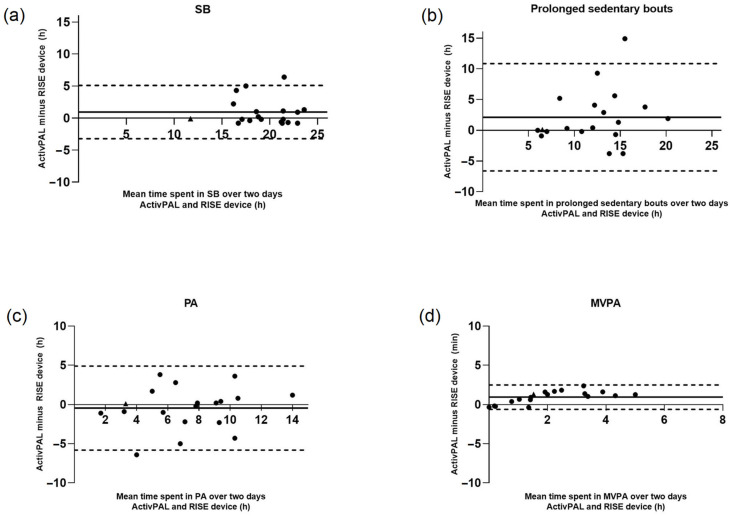
Bland–Altman plots with mean time spent in (**a**) SB, (**b**) prolonged sedentary bouts, (**c**) PA, and (**d**) MVPA over two days for the ActivPAL and RISE devices in the free-living setting. Solid line (─) represents mean difference, dashed lines (---) represent limits of agreement. Measurements are represented in hours (h). SB = sedentary behaviour, PA = physical activity, MVPA = moderate to vigorous activity. Participants with data from only one day are denoted by ▲.

**Table 1 sensors-25-03308-t001:** Participant characteristics in laboratory and free-living settings.

	Laboratory Setting (n = 25)	Free-Living Setting (n = 19)
**Age (years)**	66 ± 11.8	73 ± 10.2
**Sex (N (% female))**	8 (32)	5 (26)
**BMI (kg/m^2^)**	26.1 ± 4.2	25.7 ± 4.4
**Cause of stroke**		
** ** **Infarct (%)**	21 (84)	16 (84)
** ** **Haemorrhage (%)**	4 (16)	3 (16)
**Location**		
** ** **Left (%)**	11 (44)	13 (68.4)
** ** **Right (%)**	13 (52)	5 (26.3)
** ** **Cerebellum (%)**	1 (4)	1 (5.3)
**Time since stroke (years)**	20 [18–20]	19.5 [15–20]
**Walking aid**		
** ** **Walker (%)**	4 (16)	2 (10.5)
** ** **Crutches or cane (%)**	2 (8)	2 (10.5)
** ** **None (%)**	19 (76)	15 (79)
**BI**	20 [18–20]	19.5 [15–20]
**10 MWT (km/h)**	4.0 ± 1.1	3.6 ± 1.1
**FAC**	5 [3–5]	5 [3–5]

n = number, BMI (kg/m^2^) = Body mass index in kilograms per square meter, BI = Barthel Index score ranging from 0 to 20, 10 MWT (km/h) = 10 m walking test in kilometres per hour, FAC = Functional Ambulation Category score ranging from 0 to 5. Value notations indicate the mean ± standard deviation and median [interquartile range].

**Table 2 sensors-25-03308-t002:** Sensitivity, specificity, and PPV of the RISE device in the laboratory setting.

Movement Category	Sensitivity	Specificity	PPV
**SB**	0.93 [0.88–0.97]	0.96 [0.91–1.01]	0.95 [0.90–1.00]
**PA**	0.96 [0.91–1.01]	0.93 [0.88–0.97]	0.95 [0.93–0.98]

PPV = positive predictive value, SB = sedentary behaviour, PA = physical activity, [ ] = 95% confidence interval.

**Table 3 sensors-25-03308-t003:** Distribution of misclassification of the RISE device in the laboratory setting.

Movement Category Video	Total	Lying	Sitting	Standing	Walking	Cycling
**SB**	6.6			52.8	45.3	1.9
**PA**	1.3	5.0	95.0			

Values in this table represent the percentage of instances where the RISE device made a misclassification for each movement category based on its categorisation using video footage. The “total” column indicates the overall percentage of misclassifications made by the RISE device. Of this total percentage, the table shows the distribution of these misclassifications across different movement categories. SB = sedentary behaviour, PA = physical activity.

**Table 4 sensors-25-03308-t004:** MAPE, MPE, and mean time per movement category of the RISE device versus ActivPAL measured over two days in the free-living setting.

Movement Category	MAPE in %	MPE in %	Mean Time, RISE Device	Mean Time, ActivPAL	Mean Difference	ICC [95% CI]
**SB**	9.7	11.0	19 h and 10.8 min	19 h and 34.2 min	0 h and 57.6 min	0.8 [0.5–0.9]
**Prolonged sedentary bouts**	19.8	1.6	11 h and 4.2 min	13 h and 11.4 min	2 h and 7.2 min	0.7 [0.2–0.9]
**PA**	N.A. *	N.A. *	7 h and 27.6 min	7 h and 0.6 min	−0 h and 27 min	0.8 [0.5–0.9]
**MVPA**	N.A. *	N.A. *	34.8 min	91.8 min	0 h and 57.6 min	0.5 [−0.2–0.8]

MAPE = Mean absolute percentage error, MPE = mean percentage error, ICC = intraclass correlation coefficient, 95% CI = 95% confidence interval, SB = sedentary behaviour, N.A. = not applicable, h = hour, min = minutes, PA = physical activity, MVPA = moderate to vigorous activity, * since ≥1 value was <1, MAPE/MPE cannot be analysed.

## Data Availability

The data are available upon reasonable request.

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
