# Peer review of "Criterion and Construct Validity of the Pocket-Worn RISE Device to Assess Movement Behaviour in Community-Dwelling People with Stroke"

_sensors, 2025, doi:10.3390/s25113308_

Round 1
Reviewer 1 Report
Comments and Suggestions for Authors
Thank you for the opportunity to review this manuscript. The objective of this article was to evaluate the validity of the pocket-worn RISE device for measuring movement behavior in community-dwelling stroke patients.
I think this paper is well written and clear. The methods are reasonable, and the results are presented nicely. I think there is one limitation – a metabolic cart could have been used to assess physical activity intensity in the laboratory setting. I think this limitation should be mentioned in the limitations section of the discussion.
Introduction:
The introduction does a nice job of highlighting the importance of identifying a device that can objectively monitor sedentary behavior and physical activity. The objective is stated very clearly, and there is justification for using the ActivPal as the comparison.
Methods
The methods are clearly described.
Results
The results are clearly described.
Discussion
The discussion is clear and does a good job of summarizing the validity of the RISE device in laboratory and free-living settings.
In other papers I’ve read, people assess the validity of physical activity intensity classifications (for example MVPA) in a laboratory session with a metabolic cart. I think a limitation of this study is that no protocol was included to assess physical activity intensity validity in the laboratory setting. This should be highlighted in the limitations section of the discussion.
Author Response
Please see the attachment.
Thank you very much for taking the time to review this manuscript. We have carefully considered your comments, and the manuscript has been revised following each of the recommendations.

Reviewer 2 Report
Comments and Suggestions for Authors
Please see attached word document.

Generally the manuscript is well written, however as part of the improving the clarity of the points made this will improve the English and readability of the document.
Author Response
Please see the attachment.
Thank you very much for the extensive feedback. We are grateful for your recommendation to structure the Methods more systematically and in more detail. These changes, made in response to your review, have strengthened the methodological transparency and clarity of the manuscript.

Round 2
Reviewer 1 Report
Comments and Suggestions for Authors I do recommend accepting the paper in it's current form.Reviewer 2 Report
Comments and Suggestions for Authors
Thank you for your time to response to the comments on the initial manuscript.
You have addressed the comments raised sufficiently.